# Quantitative Phase Imaging to Study the Effect of Sodium Dodecyl Surfactant on Adherent L929 Fibroblasts on Tissue Culture Plates

**Sonthikan Sitthisang [1],[†], Jeeranan Boonruangkan [1],[†], Meng Fatt Leong [2], Kerm Sin Chian [1] and Young-Jin Kim [3],***

[1]  School of Mechanical and Aerospace Engineering, Nanyang Technological University (NTU), Singapore 639798, Singapore; sitt0002@e.ntu.edu.sg (S.S.); boon0020@e.ntu.edu.sg (J.B.); askschian@ntu.edu.sg (K.S.C.)

[2]  School of Applied Science, Temasek Polytechnic, Singapore 529757, Singapore; mfleong@tp.edu.sg

[3]  Department of Mechanical Engineering, Korea Advanced Institute of Science and Technology (KAIST), Daejeon 34141, Korea

*  Correspondence: yj.kim@kaist.ac.kr

†  These authors contributed equally to this work.

**Abstract:** Decellularization is the process of removing cellular components from native tissues or organs to obtain an acellular, collagenous scaffold for use in tissue engineering and organ regeneration. Surfactants are widely used to produce acellular scaffolds for clinical applications. However, cell–surfactants interactions have not been studied in depth. Cell-surfactant interaction was studied in a time-lapsed manner using sodium dodecyl sulfate (SDS) solution (surfactant) on adherent L929 fibroblasts as a test solution, phosphate-buffered saline (PBS) solution as control solution (isotonic), and deionized water as positive test solution (hypotonic), respectively. The QPI results show changes in the relative height and cross-sectional area of the cells, with various test solutions and exposure times. In particular, it was observed that the removal of the cell with SDS involved the disruption of the cellular membrane and detachment of the cell contents from the adhering surface. This study demonstrated the feasibility of using the QPI technique to understand the decellularization process.

**Keywords:** quantitative phase imaging; L929 fibroblasts; cell removal; decellularization

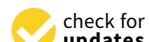



## 1. Introduction

The global shortage of donor organs has resulted in long waiting times for patients requiring tissue or organ transplants. To address this shortage, the use of tissue-engineered scaffolds to regenerate organs was studied extensively. Scaffolds are acellular structures used to support the attachment, growth and development of host cells and tissues during tissue and organ regeneration process. The scaffolds can be derived from various methods including decellularizing donor tissues or organs [1]. The scaffolds, subsequently re-populated with host's cells, are used to regenerate the organs (Figure 1a,b).

Decellularized scaffolds derived from organs offer many advantages over synthetic scaffolds as they contain inherent physical and biochemical similarities with the native extracellular matrices (ECMs) of host tissues and organs [2]. Ideal scaffolds derived from donor tissues or organs must be free of all cellular remnants to minimize the host immune response when implanted. In decellularization processes, it is important to minimize residual cellular components whilst preserving the ECMs and ECM proteins in the scaffolds [3,4].

Surfactants were used to decellularize tissues or organs, such as heart and kidney, and these scaffolds were successfully implanted in small animals [4,5]. Sodium dodecyl sulfate (SDS) is one of the most effective and commonly used surfactants to decellularize tissues and organs. Our previous study confirmed that SDS effectively decellularized porcine esophagi [6].

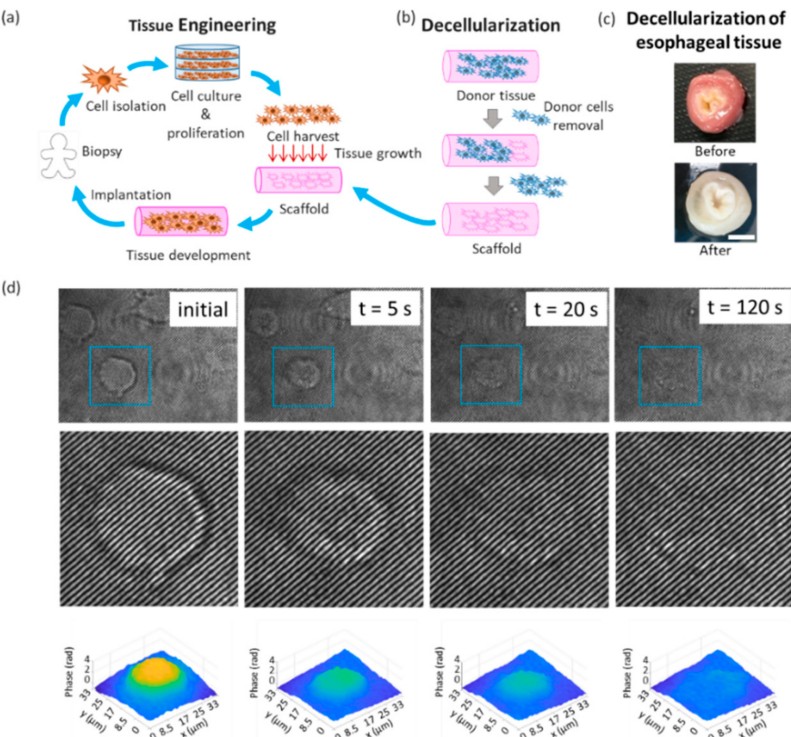

**Figure 1.** (**a**) Tissue engineering and decellularization processes. (**b**) The decellularization process of a tissue to obtain the scaffold. (**c**) Optical images of an esophagus before and after decellularization. Scale bars represent 1.0 cm. (**d**) Time-lapsed topographic phase maps showing the interaction of a L929 fibroblast with SDS at 5, 20, and 120 s.

However, the real-time interactions between SDS and cells during decellularization are not well understood, except that the process is very rapid. An in-depth understanding of SDS–cell interaction is important in providing both mechanistic and kinetic insights of the cell removal process from the native ECM. It is the hope that this study can provide both qualitative and quantitative evaluations of surfactant choices for tissue and organ decellularization. An example of a porcine esophagus before and after decellularization is shown in Figure 1c.

A quantitative phase imaging (QPI) technique is widely used for the quantitative measurements of cellular events [7–10], including the study of cellular membrane fluctuation, mass transport, and cell division [11]. In this paper, we explore the feasibility of using the QPI technique as a qualitative and quantitative tool for studying SDS–cell interactions using real-time imaging. Real-time interactions between live adherent L929 fibroblasts with phosphate-buffered saline (PBS), deionized water, and SDS solutions were studied. Unlike conventional biological imaging techniques, QPI offers the advantage that cells do not have to be fluorescently labeled, a process that poses unwanted effects such as photocytotoxicity and photobleaching [7]. This study shows that the QPI enables simultaneous imaging and the quantitative measurements of cell–surfactant interactions in real-time. QPI has an outstanding advantage in that it offers a label-free measurement, which is useful in studying surfactant-cell interactions as it does not have labelling reagents that may cause any disturbances. This study also addresses the notion that the quantitative results, e.g., peak height, cell area from QPI, are practical for the study of decellularization. Therefore, QPI is a robust technique for studying decellularization and was newly demonstrated in this paper.

Figure 1d shows an overview of the data processing in QPI. The QPI interferograms of a L929 fibroblast treated with SDS solution or control solutions were captured as a series of time-lapsed images at 0, 5, 20 and 120 s to observe the cell interactions in each timing.

Then, the QPI interferograms were converted to topographic phase maps to show their quantitative information.

## 2. Materials and Methods

### 2.1. Fibroblast Cell Preparation

We used fibroblasts in this study because of their abundance and importance in maintaining the ECM in the body. The cells are also often used in cytotoxicity testing in vitro [12–14]. L929 fibroblasts were culture-expanded in Dulbecco's Modified Eagle's Medium (DMEM, Gibco, Amarillo, TX, USA) supplemented with 10% (*v*/*v*) fetal bovine serum (FBS, Gibco, Amarillo, TX, USA), and 1% (*v*/*v*) antibiotic and antimycotic solution (Gibco, Amarillo, TX, USA). The fibroblasts were grown under standard cultured conditions of 37 °C and 5% $CO_2$. Confluent fibroblasts cultures were then harvested using 0.1% Trypsin-EDTA solution (Gibco, Amarillo, TX, USA) and sub-cultured to obtain a working population. Fibroblasts were seeded in 24-well cell culture plates (Corning, New York, NY, USA) at a density of $10^4$ cells/$cm^2$ for 2 days and maintained under standard culture conditions of 37 °C and 5% $CO_2$ prior to exposure to various test solutions.

### 2.2. SDS Solution Preparation

The SDS solution was prepared in PBS (Sigma-Aldrich, Saint Louis, MO, USA) at a concentration of 200 ppm (*w*/*v*). The SDS concentrations used in organ decellularization typically ranged between 1000 to 100,000 ppm [6,15,16]. However, at these SDS concentrations, their effects on the cells were too rapid to be imaged using our current QPI setup. In our study, we used SDS concentration of 200 ppm. Both deionised water and PBS were used as controls.

### 2.3. QPI Setup

The QPI setup in our study is shown in Figure 2a [17]. The configuration included a frequency comb (FC-1500-250-WC, M-VIS, Menlo Systems, Planegg, Germany) that was used as a speckle-tailored coherent light source for QPI to study the dynamic of cellular changes such as osmotic swelling [18]. The broad super-continuum from the frequency comb ($\lambda$ = 620–980 nm) was delivered to a band-pass filter ($\lambda$centre = 750 nm, bandwidth = 10 nm, FB750-10, Thorlabs, Newton, NJ, USA) to control the coherence for high-contrast interferograms. The laser beam was focused by a plano-convex lens to an optical diffuser (LSR-3010-6D-VIS, Optotune, Dietikon, Switzerland) to suppress the coherent speckle noise before illuminating the L929 fibroblasts. The cells, seeded on culture wells, were placed on the sample stage of an inverted microscope (IX83, Olympus, Tokyo, Japan). To image the solution–cell interaction, the culture medium was first removed and immediately replaced with either PBS (isotonic solution), deionized water (hypotonic solution), or SDS (surfactant) solution. Scattered light from the cells exposed to each solution was collected by an objective lens (Numerical aperture = 0.7, Magnification = 60, Olympus, Tokyo, Japan) before being delivered to a modified version of Michelson's type QPI unit, which satisfies both off-axis and common-path methods [19]. A charge-coupled device (CCD) camera (Thorlabs) was used to capture time-series interferograms of the cell at 12 fps (frame per second). QPI exploits optical phase delay (OPD) altered by the laser beam passing through the sample to collect quantitative data. The relationship between OPD (($\Delta\varphi$(x,y)) and the illumination wavelength ($\lambda$) (nm), the refractive indices of cell ($n_c$), refractive indices of the surrounding medium the cell ($n_m$), and the cell thickness (h(x,y)) (nm), is shown in Equation (1) below [20]:

$$\Delta\varphi\,(\text{x,y}) = (2\pi/\lambda)\cdot(n_c - n_m)\cdot h(\text{x,y}) \tag{1}$$

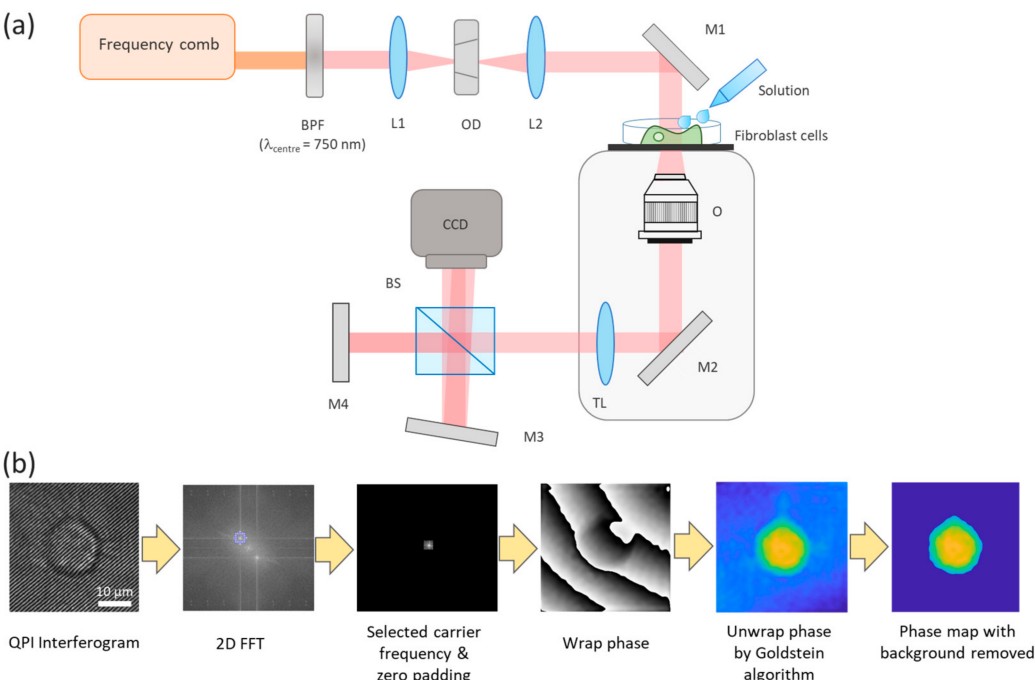

**Figure 2.** (**a**) Experimental QPI setup. (**b**) Phase extraction, unwrapping and background removal from a QPI interferogram. BPF: band-pass filter, L1: plano-convex lens 1, OD: optical diffuser, L2: plano-convex lens 2, M1: mirror 1, O: objective lens, M2: mirror 2, TL: tube lens, BS: beam splitter, M3: mirror 3, M4: mirror 4, and CCD: charge-coupled device. Scale bar is 10 μm.

To retrieve the collected OPD, the following analytical techniques were applied. Firstly, a Fourier transform was performed on the QPI interferograms to spatially filter out the carrier frequency of the optical field. Secondly, the Goldstein's phase unwrapping technique was applied to acquire the correct phase value from the $2\pi$ phase ambiguity [21]. Finally, the OPD over the field-of-view result can be shown as an optical phase map or topographic phase map of the cell. Figure 2b shows a sample interferogram of a single cell of ~20 μm-diameter and its resulting phase maps, with and without background. The background of the cross-sectional area was removed to determine the total area of the cell for subsequent analysis. The cross-sectional area of a cell is related to the integrated OPD which varies with the changes in the thickness, the refractive indices of the cell and/or the surrounding medium.

*2.4. Observation of Solution–Cell Interactions*

To evaluate the solution–cell interactions over time, 3 fibroblasts were imaged to determine their peak heights and cross-sectional areas of the cells at pre-determined intervals (at t = 5, 20, 60 and 120 s) and compared to the untreated cells at t = 0 s (i.e., before replacing the culture medium with the test solution). The ratio $Y_t/Y_0$, was used to determine the measured parameters, namely peak height and cross-sectional area of the cell, where $Y_0$ and $Y_t$ were the measured parameters at time t = 0 and t seconds. In the case of cell peak heights, "$Y_0$" and "$Y_t$" correspond to the maximum values obtained from the OPD (rad) of the phase map at the respective times. The cross-sectional area of the cell was calculated from the cross-sectional size of a cell over the field-of-view ($\mu m^2$) at the respective times.

**3. Results and Discussions**

The results of the QPI study on the interaction between PBS, deionized water, and surfactants with L929 fibroblasts are shown in Figure 3. Figure 3a shows the interferograms pattern generated on the cells; whereas Figure 3b, derived from the interferograms, shows the topographic phase map to indicate the peak height and cross-sectional area of the cell.

The colors on the phase map indicate the measured OPD expressed in radians (rad). Finally, in Figure 4, the relative change in cell peak height and the cross-sectional area of the cell are plotted against the exposure time in PBS, deionized water and SDS.

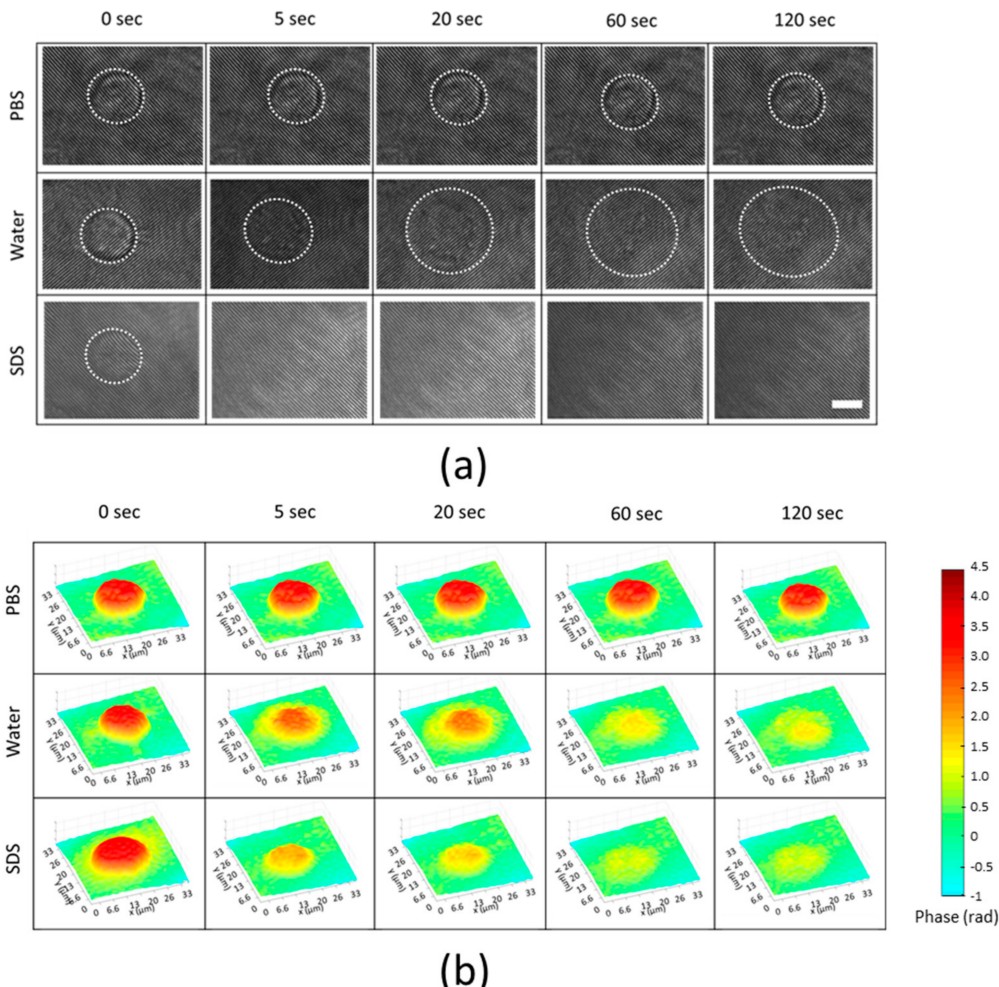

**Figure 3.** (**a**) Interferograms of L929 fibroblasts in PBS, deionised water, and SDS over time. Scale bar is 10 μm. (**b**) Topographic phase maps showing the peak height and cross-sectional area of L929 fibroblasts exposed to PBS, deionized water and SDS over time.

The cells exposed to PBS showed little to no change with time, as shown in the interferograms in Figure 3a-PBS and the topographic phase maps in Figure 3b-PBS. This is evident in Figure 4a,b where no relative changes were observed. The results obtained for PBS were expected because, being isotonic, it had the same osmolarity as the cell and the diffusion processes did not affect the integrity of the cell wall.

The transportation of water across the cell membrane via the phospholipid bilayers and aquaporins is well documented [22]. In Figure 3a, the cells exposed to deionized water up to t = 20 s, showed a simultaneous increase in the cross-sectional area but a rapid decrease in the cell peak height. The observed changes are likely to be caused by the diffusion of water due to the higher concentration of the cell cytoplasm. It is known that mammalian cells do not have rigid cell walls, and an increase in the cell volume can result in the flattening of the cell structure [23].

However, at exposure times of t > 20 s, as shown in Figure 3a, no cell walls were visible on the interferograms. The topographical phase maps of these interferograms (Figure 3b) also show a very significant decrease in the cell peak height. The combined observations of (i) a significant reduction in cell peak height and (ii) sudden loss of cell wall images in the interferograms, strongly suggested that the disruption in the cell membrane resulted in

the loss of cytoplasmic and nucleic matters; i.e., the basis of the decellularization process. The cells placed in a hypotonic solution take in water across their membrane. The uptake of excess water can produce enough pressure to induce the cytolysis or rupturing of the cell. It was observed from the topographical images that cell residues remained adhered to the substrate, as shown by the yellow-colored region. The results in Figure 4a,b show that after t > 60 s, both the relative cell peak heights and cross-sectional areas remained constant, further suggesting the collapse of the cell membranes. The higher final cell peak height suggested higher levels of residues remaining in the cells.

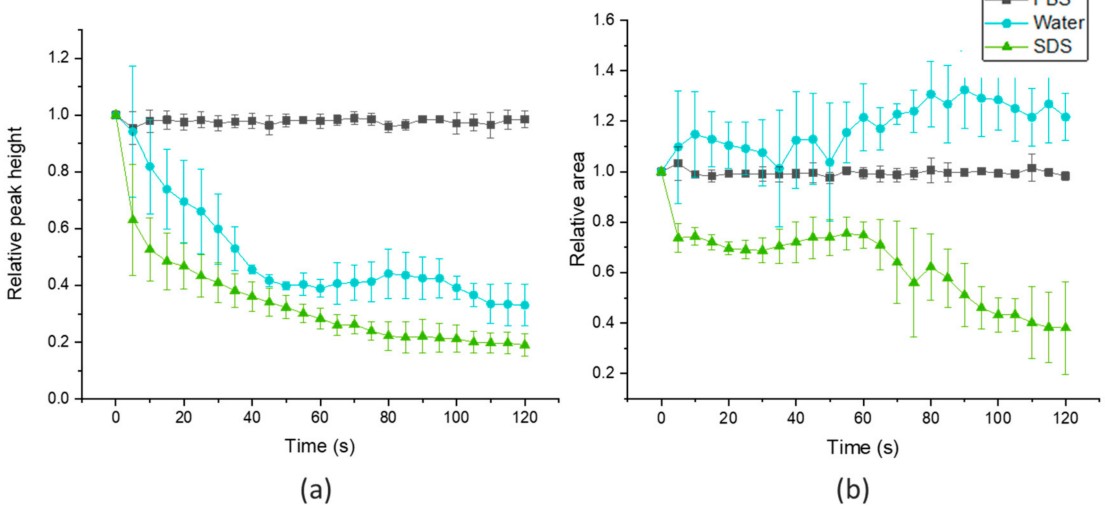

**Figure 4.** Real-time QPI data showing the relative change in (**a**) peak height and (**b**) cell cross-sectional area of the L929 fibroblast cells over time in PBS, water, and SDS (number of fibroblast cells = 3).

Our QPI results show that both deionized water and SDS caused lysis to the cells by different mechanisms. The cells treated in deionized water were lysed by the pressure increase in the cells due to diffusion of water into the cell. However, the remnants of the lysed cell remained on the substrate. In the case of the SDS-treated cells, the disruptions to the cell membranes were significantly more rapid than those in deionized water. According to Figure 4a, the relative cell peak heights of the cell with SDS treatment decreased much faster than the cell with deionized water treatment. The decrease in the relative peak height indicates the cell lysis, which is the breaking down of the cell membrane and a loss of the cellular components. The rate of the cell lysis was calculated by the difference in relative cell peak heights per second. The rate of the cell lysis of PBS treatment was nearly zero $s^{-1}$ and it indicated that the cells did not lyse in PBS. The results found that the SDS provided a much higher rate of cell lysis ($0.07 \ s^{-1}$) than the rate of cell lysis by water ($0.01 \ s^{-1}$) in first 5 s. The results show that the cell was lysed faster in SDS than in water.

More importantly, the topographical phase maps show a more efficient removal of cell residues. Unlike cells treated in deionised water, the interaction between SDS solution and cell membranes was chemical in nature. The process involved the chemical solubilsation of the hydrophic bilipid layer on the cell membrance by the SDS moleules. This solubilising process results in the rapid dissolution of the bilipid layers of the cell membrane. The rapid rate of the chemical solubilisation of the cells with SDS could only be studied and imaged using the QPI described in this paper.

Compared with conventional biological imaging techniques, our quantitative QPI technology provided real-time cell–solution interactions which allowed an estimation of the cell disruption rate. Thus, our set-up has the potential for future surfactant screening for decellularization applications within minutes, which are much faster than decellularization that requires days to weeks.

## 4. Conclusions

In conclusion, QPI provided us with a clear understanding of the effects of PBS, deionized water, and SDS on L929 fibroblasts. Our QPI study for cell–PBS interaction reliably showed that the results obtained were consistent with the expected outcomes of the cells in isotonic solutions, such as PBS and deionized water. The application of QPI to study the interactions of L929 fibroblasts in deionized water and SDS provided important real-time visual and qualitative comparisons on the rate of cell lysis and the extent of the removal of cell remnants from the substrate, which would otherwise require additional examinations and analyses.

**Author Contributions:** Conceptualization, K.S.C. and Y.-J.K.; methodology S.S. and J.B.; Cell preparation S.S.; QPI setup J.B.; Data analysis S.S., M.F.L. and J.B.; Original draft preparation, S.S. and J.B. All authors participated in preparatory discussions and manuscript improvement. All authors have read and agreed to the published version of the manuscript.

**Funding:** This work was partially supported by the National Research Foundation of the Republic of Korea (NRF-2012R1A3A1050386, NRF-2020R1A2C2102338, NRF-2021R1A4A1031660); KAIST UP Program; Singapore National Research Foundation (NRF-NRFF2015-02).

**Institutional Review Board Statement:** Not applicable.

**Informed Consent Statement:** Not applicable.

**Data Availability Statement:** Not applicable.

**Acknowledgments:** The authors thank Seng-Woo Kim for useful discussions and insightful suggestions.

**Conflicts of Interest:** The authors declare no conflict of interest.

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
