# Peer review of "Quantitative Phase Imaging to Study the Effect of Sodium Dodecyl Surfactant on Adherent L929 Fibroblasts on Tissue Culture Plates"

_photonics, doi:10.3390/photonics8110508_

Round 1

Reviewer 1 Report

The review paper “Quantitative Phase Imaging to Study the Effect of Sodium Dodecyl Surfactant on Adherent L929 Fibroblasts on Tissue Culture Plates” by Sonthikan Sitthisang  et.al., summarizes real-time cell-surfactant interactions using quantitative phase imaging (QPI) technique. Adherent L929 fibroblasts interactions with  phosphate buffered saline (PBS), deionized water and sodium dodecyl sulfate (SDS) on culture plates were studied.

The concept and idea of the work is good, but QPI is an already well-established technique, so novelty of this work is missing. This manuscript needs major revisions, only then I can make a decision.

  1. Abstract Line 15-16, “cell-surfactant interaction by time-lapsed of 3 test solutions on cells using the quantitative phase imaging”. Kindly paraphrase this. The sentence does not make any sense.

It should be this way instead: cell-surfactant interaction by time-lapsed using sodium dodecyl sulfate (SDS) solution (hypertonic), on adherent L929 fibroblasts as test solution, phosphate buffered saline (PBS) solution as control solution (isotonic) and deionized water as positive test solution (hypotonic), respectively.

  1. Line 39-41, “ In decellularization processes, ……………………... proteins in the scaffolds. Kindly add references to support this statement.
  2. Figure 1 (c). Scale missing?
  3. Figure 1 (d). Kindly present it in a way that words/texts can be seen. I had a hard time reading.

Kindly understand that collecting good data does not “qualify” for an acceptable paper/article. Drafting the manuscript in a readable way is very important for a good paper. Kindly put some time to re-address all the figures in a presentable manner rather than just throwing in a bunch of figures which cannot be understood at all. This article has a lot of good data, that needs to be addressed and presented in a readable format.

  1. Kindly add quantitative cell peak height if possible? Would be a good way to compare rate of cell lysis.
  2. Figure 1 (d) and 3 (b). Kindly present the phase maps better. Also, quantitative information from these maps should be presented better.
  3. Why was this study conducted only for 120 seconds?
  4. Figure 3 (a) Kindly present the optical images of cells better, if possible, outline the cellular area.
  5. Another question I have is about Repeatability. QPI system is well established. However, with how many cells was the study conducted, why no standard deviation with series of cells is presented?

Reviewer 2 Report

Straightforward paper, with convincing results. However, there are a few questions that should be addressed prior to publication. 

1) It is not clear how the authors are getting a holographic image - please clarify.

2) The results cells in tissue culture dishes, as opposed to actually decellularized tissues. The authors have not shown a single scaffold. This is critical to support their contention. The authors need to image cells on actual scaffolds.

Reviewer 3 Report

The submitted manuscript is interesting because it is related to the use of the QPI technique to analyse the interaction of cells with 3 types of surfactants, which may contribute to the understanding of the decellularization process. However, the manuscript needs corrections before publication. In my opinion, the presented analysis is rather superficial considering the possibility to retrieve from QPI data many physical parameters quantitatively characterising cells and their changes induced by environmental factors e.g. osmotic stress. Detailed comments are provided below.

Major comments

1) From the introduction but also the remaining sections of the article, it is difficult to understand what is the authors' new contribution in the field of quantitative phase imaging. The introduction should be revised to indicate the advantages of QPI and whether the use of QPI to study the interaction of cells with surfactants and the effect of different centres on cell features.

In recent years, many papers have been published related to QPI and especially to digital holographic microscopy and digital holographic tomography, which have been used to study various physical parameters of cells (profile, volume, dry mass, dry mass density, integral and average refractive index of cells, the refractive index of cell organelles, etc.) and their interaction with the different media in which they are located.  Here are some examples:

https://doi.org/10.1038/s41467-018-08207-5

https://doi.org/10.1364/BOE.8.002222,

https://doi.org/10.1117/1.3290242

https://doi.org/10.1117/1.2804926

https://doi.org/10.1002/cyto.a.23082

Why didn't the authors decide to perform more complex analysis of the cell parameters that we can obtain from QPI and the knowledge of the refractive index of the medium? This would have greatly enriched the content of the manuscript. Particularly, since the influence of osmotic stress on the eukaryotic cells ( its refractive index and morphology) was already indicated in a much more complex manner. See examples:

https://doi.org/10.1364/OPEX.13.009361

https://doi.org/10.1002/cyto.a.23082

https://doi.org/10.1038/s41467-018-08207-5

https://doi.org/10.3390/cells8111368

In my opinion, the analysis of the results has been carried out in a somewhat cursory manner and should be complemented by an analysis of additional parameters retrieved from the QPI data.

2) Have the authors done any comparative studies e.g. using conventional imaging techniques: fluorescence microscopy, confocal microscopy, phase contrast, DIC? This would greatly facilitate the comparison of results and demonstration of the superiority of QPI over other common imaging modalities.

3) To ensure the completeness of the results, the authors should include results representing the different steps leading to the final integrated phase distribution, i.e. an example of the recorded digital hologram, its Fourier spectrum before and after filtering, etc.

4) Did the authors analyse the effect of the partial penetration of the cell wall by the surrounding medium on the changes of integrated phase distribution? How the OPD of the cell wall was changed over time? How the cell wall was distinguished? What criterion was used for this purpose?

5) How were the refractive indices of the media in which the cells are located measured? Please describe this and include information on the refractive index values of each medium. Did the authors

6) In lines 122-123 it was stated that: "The background of the cross-sectional area was removed to determine the total area of the cell for subsequent analysis.”- explain the used segmentation algorithm?

Minor comments

1) Please provide the spectral ranges of the frequency comb, which was used as the light source.

2) The circuit description is missing some abbreviations referring to the optical components in the schematic (Fig.2a).  Please explain/insert all abbreviations in the text or the figure caption.

3) Please add the basic data from the technical specifications of all the optical elements used in the QPI system, because in the current version the authors have introduced the specifications of only some elements. Please also provide the manufacturer of these components.

4)Please correct the sentence in lines 110-112 as it is unclear.

5) Section two is badly formatted. The text should be aligned to both margins, not just the left. Please correct this.

6) Figure 2 should be closer to the description in the text. The components of the optical system are described on a different page than its schema, which may be confusing for the reader. Please move this figure to the same page as the system description, or include an explanation of the designations of the individual elements in the figure caption.

7) There are typos in the article for example in the figure caption: " Optical mages of an esophagus before and after decellularization". The authors should correct such errors

Round 2

Reviewer 1 Report

N/A

Reviewer 3 Report

The manuscript can be published in recent form.